# Impact of COVID-19 on Substance Use Disorder Treatment: Examining the Influence of In-Person and Telehealth Intervention on Outcomes Using Real-World Data

**DOI:** 10.3390/healthcare13010084

**Published:** 2025-01-06

**Authors:** Cinta Mancheño-Velasco, Marta Narváez-Camargo, Daniel Dacosta-Sánchez, Ana de la Rosa-Cáceres, Óscar M. Lozano

**Affiliations:** 1Department of Clinical and Experimental Psychology, University of Huelva, 21071 Huelva, Spain; cinta.mancheno@dpces.uhu.es (C.M.-V.); marta.narvaez@dpces.uhu.es (M.N.-C.); daniel.daco@dpces.uhu.es (D.D.-S.); anamaria.delarosacaceres@unir.net (A.d.l.R.-C.); 2Natural Resources, Health and Environment, University of Huelva, 21071 Huelva, Spain

**Keywords:** substance use, hybrid treatments, outcomes, retention, COVID-19

## Abstract

**Background**: The COVID-19 health crisis challenged healthcare systems around the world, leading to restrictions in access to face-to-face healthcare services, and forcing rapid adaptation to telehealth services. At present, there is a gap in the functioning of this adaptation in drug-dependence centres. The present study analyses, over four years, care indicators on the care modality (face-to-face vs. hybrid), the patient profile and the impact on retention in treatment. **Methods**: Retrospective observational study with data collected between 14 March 2019 and 21 June 2023. The electronic health records of 44,930 patients were analysed according to different moments and selected based on the different health measures imposed by the COVID-19 pandemic. Patients were classified according to whether they received an in-person or hybrid intervention. Bivariate statistics and logistic regression analysis were applied. **Results**: The trend over time shows an increase in the number of patients seen in addiction centres. However, no notable changes within the in-person care modality and a modest increase in telehealth services are observed. Telehealth is primarily used among patients with opiate addiction, as well as with those with comorbid mental disorders. Logistic regression analysis shows that patients in a hybrid modality are more likely to remain in treatment. **Conclusions**: Results show that hybrid care is associated with higher patient retention rates. Despite this, different profiles are mostly treated with in-person interventions rather than hybrid modalities. Future studies should explore how to generalise personalised hybrid care among SUD patients considering factors such as patients’ educational level, employment status or accessibility to mental health services.

## 1. Introduction

The COVID-19 pandemic has led governments in different countries to adopt measures that impact the lives of citizens. In terms of health, the pandemic represented a trigger for promoting telehealth [1]. This can be understood as the incorporation of technological means in the provision of clinical services, such as remote patient monitoring, videoconferencing between therapists and patients in routine clinical practice, and mobile applications [2].

In the field of the treatment of patients diagnosed with substance use disorder (SUD), although telehealth activities were already available before the pandemic [3,4], after its onset, the number of protocols for incorporating telehealth into clinical practice increased, which has probably influenced the exponential expansion of these practices in addiction centres [5]. After several years of telehealth expansion, several authors have highlighted the benefits of this care modality, such as the possibility of having a more flexible and agile care resource without the need to commute to care centres [6]. However, it has also been noted that this modality makes patient–therapist interaction difficult, which can be a handicap for treatment [7]. Further, not all treatment services can be remotely implemented; security and privacy issues may cause patients to feel a bit uncomfortable; internet access may be restricted in some areas; or there may be limited availability of patient’s medical history to healthcare providers. This means that telehealth services cannot be provided on an equal basis to all patients.

Different studies have been conducted during specific phases of the COVID-19 pandemic with SUD patients. For example, reviews such as that of Mark et al. [8] indicate that the results of in-person and telehealth treatments are similar. Subsequent studies have shown that the application of telehealth techniques can be beneficial in the treatment of patients with SUD. For example, Sistad et al. [9] reported that telehealth resources can have a positive effect on the earlier stage of treatment in patients with SUD. Similarly, Gainer et al. [10] reported that the application of telehealth techniques at the beginning of treatment in patients with SUD decreases the likelihood of dropout compared with those who only receive in-person treatment. In the case of opioid-dependence patients, different studies have suggested that telehealth treatments match and even increase retention rates of patients receiving medication for opioid use disorders [11,12,13,14]. Further evidence from SUD patients from other studies suggests that the quality of life of patients who receive both telehealth and in-person treatments is equivalent to that of patients who complete an exclusive telehealth intervention [15,16]. Gliske et al. [17] conducted a retrospective study of 3642 SUD patients, classifying them according to whether they received an in-person, hybrid or telehealth treatment. Three months after the start of treatment, these authors found no notable differences among the three groups, highlighting that those patients following a hybrid program reported better levels of general health than patients receiving telehealth assistance did. After one year of follow-up, the results indicated that those following the hybrid intervention had greater retention and a lower proportion dropped out of treatment than those who participated in traditional (in-person) or online-only programmes (telehealth) [18].

Overall, the above studies suggest that, compared with in-person treatment, the use of telehealth techniques led to results similar to those of in-person treatment on indicators associated with dropout, retention in treatment or quality of life, and that the adoption of hybrid treatments may be even more positive. However, most of these studies were conducted during specific time periods, and their findings may have been influenced by the temporary legal measures adopted by governments to mitigate the impact of the COVID-19 pandemic. In turn, all of the above studies have been conducted in the US. Therefore, it can be useful to provide evidence over an extended period, in countries other than the US, to evaluate how telehealth services have been implemented in addiction treatment centres. Specifically, how have telehealth services evolved from the restrictions imposed by the COVID-19 pandemic to their lifting, compared to face-to-face care? Additionally, what impact do they have on patient retention?

Considering that the legal measures for drug dependence centres have changed during different phases of the pandemic, the present study aims to analyse across four different periods (pre-pandemic, state of alarm, controlled recovery phase, and restrictions phase-out) in Spain, a country that implemented health-related legal measures similar to other European countries, to determine the impact of telehealth activities on patients’ retention in treatment. In this regard, several specific aims are considered: (i) to analyse the temporal evolution of in-person and hybrid treatment activity and the profile of patients adhering to each modality, and (ii) to analyse whether the hybrid modality improves retention in treatment compared with the in-person modality in different time periods.

## 2. Materials and Methods

### 2.1. Data Source

The study was conducted using the information available in the electronic health records (EHR) of 44,930 outpatients. Patients received treatment for their addiction in one of the 110 centres of the Public and Subsidised Network of Addiction Centres of Andalusia (a region of Spain with more than 8,000,000 inhabitants).

Patient EHRs from all care centres are centralised in a single database, allowing no duplication of data for each patient. It also includes automatic checks to detect inconsistencies in the data entered. The information on the EHR is based on the Treatment Demand Indicator (TDI) Standard Protocol 3.0 of the European Monitoring Centre for Drugs and Drug Abuse [19]. Each patient’s data included sociodemographic information, patterns of consumption, previous treatments and illnesses. It also contains information about their treatment and therapeutic process, clinical observations and outcomes.

The registration of the data and storage of the information complied with the General Health Law, and Ethical approval was granted by the Research Ethics Committee of the Andalusian Ministry of Health, which has certified compliance with the requirements for the ethical handling of the information.

The study received ethical approval from the Research Ethics Committee of the Ministry of Health in Andalusia (Cod. 0661-N-22), ensuring ethical standards for data management.

### 2.2. Design

Retrospective observational study based on data from patients receiving treatment between 14 March and 21 June 2022. A follow-up period of one year was included for these patients until 22 June 2023.

### 2.3. Participants

Patients in the study were undergoing treatment or started treatment during the dates specified above. Patients who met the following criteria were included: (a) had been diagnosed with opiate, stimulant (excluding patients with nicotine dependence only), hypnosedatives, hallucinogens, cannabis or alcohol dependence; (b) had attended at least one treatment information session and at least one therapy session. Exclusion criteria were: (a) patients with only behavioural addictions (i.e., pathological gambling); (b) patients who only attended the first information session of treatment.

Although COVID-19 was identified in December 2019, in many European countries, such as Spain, the legal measures associated with restricting population movements and sanitary services were not adopted until 2020. Since then, laws have been passed that have impacted the care of patients in treatment for addiction [20,21,22]. In this regard, the period was divided into four phases: (1) pre-pandemic (14 March 2019 to 13 March 2020; no restrictions on patient care); (2) the first state of alarm and extensions (14 March to 21 June 2020; home confinement of patients. The possibility of going to addiction centres with strong restrictions of capacity in the centres, spatial limitations and use of personal protective measures); (3) controlled recovery phase (22 June 2020 to 21 June 2021; the capacity of addiction centres is increased for patient care. The use of personal protective measures is maintained); (4) restrictions phase-out period (22 June 2021 to 21 June 2022; sanitary restrictions are eliminated).

During these phases, all patients were informed that appointments could be either in-person or conducted through telehealth modalities, including computerised assessments, telephone-based recovery, and telephone-based therapy. Patients could accept or decline treatment including telehealth modalities. They could also indicate which telehealth modality they preferred at any given time and could make use of these at any time during treatment.

Among the 44,930 outpatients, 80.1% were male. The mean age of the patients was 40.36 years (SD = 12.83). A total of 31.6% of the patients had alcohol dependence, 28.2% had cocaine dependence, 17.1% had cannabis dependence, 19.4% of the patients had opioid dependence, and 1.9% had hypnosedative dependence. The analysis of psychiatric comorbidity shows that 13.7% of the patients had some comorbid mental disorder not induced by drug use (1.8% psychotic disorders, 3.3% mood disorders, 6.1% anxiety disorders and 4.8% personality disorders). Of these patients, 39.2% received healthcare in a coordinated manner from the mental health services.

### 2.4. Measures

Different variables were analysed in this study. First, the treatment modality. Each patient was categorised according to whether they received in-person or hybrid (in-person and telehealth) treatment in each period analysed. That is, patients with in-person treatment were those who attended the treatment centres to receive treatment. Patients categorised as hybrid had in-person treatment and at least one telehealth clinical activity.

Second, sociodemographic variables (age, gender), variables associated with substance use (abuse/dependence disorders of alcohol, cocaine, opiates, and cannabis use) and variables associated with other comorbid mental disorders (any psychotic disorders, mood disorders, anxiety disorders, and personality disorders) and their treatment (patients in mental health services also) were studied as covariates.

The dependent variables used were retention and dropout. A patient was considered to have dropped out of treatment when a period of at least six months without receiving treatment, against medical and psychological prescriptions, was observed. The time of treatment drop-out was coded as the month following the last appointment attended by the patient.

### 2.5. Data Analysis

For the analysis of the first specific aim, uni- and bivariate statistics were applied. The chi-square test was applied to determine the associations between groups. Given the large sample size, the Phi effect size was calculated, considering the presence of at least a weak effect size when phi > 0.20 [23].

For the second specific aim, hierarchical logistic regression analyses were applied. The overall fit of the model was tested via the omnibus test, which revealed that the model is significant and has a good fit. The R^2^ of each block was also calculated to observe the increases in variance explained with the introduction of the variables.

All analyses were performed with SPSS 29.0.

## 3. Results

### 3.1. Trends in the Modality of Care and Patient Profile

During the study period, a total of 33,397 (74.3%) outpatients received in-person care. There were 11,533 (25.7%) outpatients who received hybrid treatment. However, as shown in Table 1, the two care modalities differ in each trimester.

Patients who receive in-person treatment attend most of the scheduled appointments with the therapists at the centre. Between the “pre-lockdown” and “restrictions phase out” periods, there was an increase in the number of patients seen. A significant decrease was only observed during the “first state of alarm and extensions” period. On the other hand, the average number of appointments per patient remained similar throughout the trimesters, except during the “first state of alarm and extensions” period. Among patients who have received hybrid care, several aspects can be observed. The last months of the study revealed that the number of patients per trimester has increased compared with that in the pre-pandemic quarters, with the highest value occurring during the “first state of alarm and extensions” period.

The clinical activity of these patients shows that they had fewer in-person appointments and, in addition, missed more appointments with professionals in the therapy centres. In contrast, these patients attended almost all the telehealth activities. During the “first state of alarm and extensions” and the quarter after (“controlled recovery phase”), the number of telehealth activities exceeded the number of in-person appointments.

Table 2 shows the percentage of patients in each care modality during the different phases of the study. Although the above table shows that the total number of patients within the hybrid care modality increased in the last trimester, in percentual terms, these patients represented a lower proportion than those in the pre-pandemic stage.

The profile of the patients revealed that, except for those in the “first state of alarm and extensions” period, those who received in-person care were mainly women, patients with alcohol dependence, cocaine dependence and cannabis dependence. In contrast, patients with opiate dependence are more likely to use hybrid care, regardless of the phase analysed. The treatment of patients with mental disorders varies across the phases studied, and according to mental disorders. On the other hand, it is noteworthy that, during the “first state of alarm and extensions” period only a greater percentage of patients with cannabis dependence received in-person treatment. The rest of the groups followed a hybrid modality.

### 3.2. Relation Between Treatment Modality and Treatment Retention

Table 3 shows the associations between the different study variables and treatment retention for each study period. The socio-demographic variables have scarce predictive capacity for treatment maintenance or dropout. When variables associated with dependence are introduced, the R^2^ values clearly increase. The highest odds ratios are observed in the group of patients with opiate dependence, reflecting the fact that these patients are the ones most likely to remain in treatment. The analysis of comorbid mental disorders (any psychotic, mood, anxiety, or personality disorders) shows that the presence of comorbid disorders is related to a greater probability of remaining in treatment. However, the increase in the variance explained with respect to the block represents approximately 2% in the different stages of the study. Finally, when the type of therapeutic modality is introduced, patients following a hybrid modality have a greater probability of being retained in treatment. Particularly noteworthy, in the last period analysed (“restrictions phase-out”), the increase in R^2^ was close to 6.8%.

## 4. Discussion

The present study aimed to analyse the profiles of patients receiving either in-person or hybrid care and to determine its impact on treatment retention. Since the COVID-19 pandemic, various health measures have been implemented, progressively influencing the adoption and integration of telehealth. Therefore, unlike existing evidence, the present study included a 3-year period analysis (and one more follow-up), during which various health measures were introduced in addiction treatment centres.

With respect to the first aim, the results show that the number of patients treated increased following the pandemic restrictions, compared with the pre-pandemic period. This increase was observed among patients receiving either in-person or hybrid care. This result is consistent with existing reviews of alcohol and other drug services (AODS) utilisation, which report a decline in AODS utilisation during the first alarm period followed by a gradual increase in AODS utilisation thereafter [24,25]. Additionally, the proportion of in-person appointments remained stable throughout the study period, whereas the share of telehealth-related clinical activity increased. These findings reflect the efforts made by the staff of addiction treatment centres, and the rapid organisational change that occurred following the COVID-19 pandemic, a phenomenon also noted by other authors [26].

With respect to the profile of the patients throughout the different periods, except for the first state of alarm and extension period, the profile of patients receiving in-person care or telehealth services remained relatively stable over time. However, the results revealed an increase in the percentage of women, patients with alcohol use disorders, and patients with cannabis use disorders, who received in-person care without any telehealth activity. The increase in women could be unexpected, although other authors also find similar evidence [26], as the literature usually shows that women have more difficulties attending to treatment [27,28]. In that sense, it could be expected that women would utilise telehealth support more frequently than addiction treatment centres in person.

With respect to cannabis use disorder patients, previous studies have shown that interventions based on telehealth have positive outcomes in terms of satisfaction with and motivation for treatment [29,30]. Therefore, these patients are expected to be more heavily represented in the hybrid treatment group.

On the other hand, patients with opioid use disorders are clearly the most involved in telehealth activities. This could be because telehealth activity is related to follow-up on medication for opioid use disorder [12].

For patients with comorbid mental disorders, some studies have shown that telehealth services can be comparable to in-person care [31] and may even improve their retention in treatment [32]. The present study shows that, on average, these patients receive more hybrid care across all periods. Additionally, nearly 40% of these patients receive coordinated care within mental health services. Therefore, these patients were managed by two care networks (addiction centres and mental health services). Previous studies have shown that patients with comorbid mental disorders may experience care overload and subsequently drop out of one of the care networks [33]. As a result, the adoption of telehealth measures for these patients could be an effective strategy.

Concerning the second aim, it was observed that different variables were associated with an increased probability of retention in treatment versus dropout. In addition, the use of a hybrid model and the diagnosis of opioid use disorder were among the variables most strongly associated with treatment retention. This finding is relevant in the context of the treatment of patients with SUD, as there is wide heterogeneity in dropout rates and generally low retention in treatment [34]. Therefore, in line with previous review studies [35,36], this result shows that telehealth care activity in combination with in-person care could improve patient retention. However, before considering telehealth as one of the most important factors in improving the retention of SUD patients, several factors should be taken into account. In this context, it is likely that factors such as educational level, employment, or availability of mental health facilities [37,38,39] influenced the use of a hybrid or in-person model. Therefore, it is likely that there is an interplay of personal factors combined with telehealth measures that improve retention [40].

The results of this study extend the evidence for the benefits of hybrid care beyond the periods directly affected by legal measure associated with the COVID-19 pandemic. This finding, along with evidence from other studies (i.e., [35,36]), should be considered by managers and policymakers to encourage the development of hybrid interventions so that hybrid care becomes the norm rather than the exception. In this regard, research should be promoted to optimise therapies with in-person sessions and telemedicine together. That is, when in-person assessments may be more appropriate and when telemedicine clinical sessions are more suitable. At the same time, professionals need to be trained to ensure that patients feel comfortable with telemedicine sessions. Additionally, it is crucial for drug addiction centres to consider the resources available to each patient so that treatment is adapted to their individual needs.

Despite the favourable evidence supporting hybrid care, it is important to consider some limitations to accurately interpret the results. On the one hand, the study covers a period of over three years. During this time, patients were classified as receiving in-person or hybrid care in each phase of the study. Therefore, a patient whose treatment spans different phases of the study may have received in-person care in one phase and hybrid care in another phase. While this may be considered a limitation, we believe that this analytical approach is realistic in terms of the treatment patients receive, as the analyses for each phase are such that the impact is limited. Also, it is necessary to consider that the use of telehealth is not standardised among patients. As a study conducted with real data, the application of telehealth activities depends on the resources of each patient. It is therefore important to acknowledge that these other variables may also be influencing retention and that the increased retention in treatment is not exclusively due to following a hybrid intervention modality.

On the other hand, there are different criteria for determining when a patient has dropped out of treatment or remains in treatment [41,42]. In this study, we have chosen to define treatment drop-out after a period of six months without attending appointments before clinical recommendation. During this period of time, patients may feel well and believe that they have overcome their addiction problems and could therefore stop treatment. It may therefore be appropriate to combine this outcome indicator with other quality-of-life measures. Nevertheless, studies consistently show that patients who drop out of treatment before medical recommendations generally require more therapeutic support due to relapses [43,44].

Despite the above limitations, we consider that the evidence obtained is highly valuable, as it shows that these activities improve retention in the treatment of patients with SUD, which is one of the main problems associated with the treatment of these patients. Future research could explore how to tailor telehealth services to patients’ characteristics. Beyond achieving interventions adapted to the socio-demographic characteristics of patients, research can aim to ensure that the telehealth service respects patients’ privacy, adapting to the schedules and resources available to each patient. It is likely that the greater the comfort patients feel with undergoing treatment, the greater the success of the treatment.

## 5. Conclusions

Several studies have shown that a hybrid intervention model combining telehealth and in-person sessions improves retention and reduces dropout rates among patients with substance use disorders (SUD). This study contributes to this body of research, utilising a sample of 44,930 outpatients, making it one of the largest studies conducted to date. Furthermore, the evidence obtained is naturalistic, so the results should be considered by healthcare managers and policymakers as supporting evidence for the implementation of a hybrid care model for SUD patients in real-world settings.

However, this study also raises several important research questions that need to be addressed. For example, it is crucial to investigate why patients tend to prioritise in-person care over a hybrid model, whether this preference is linked to dispositional variables (e.g., motivation) that facilitate treatment adherence, what characteristics influence the choice of one modality over the other, and how to develop a hybrid model that ensures equal access to treatment and outcomes for all patients, regardless of their resources.

## Figures and Tables

**Table 1 healthcare-13-00084-t001:** Quarterly evolution of attendance activity.

			In-Person			Hybrid				
		Patients (*n*)	In-Person Visits (*n*)	Avg. of Appointments (SD)	Attendance (Mean of % (SD))	Hybrid Visits (*n*)	Average of In-Person Visits (SD)	Attendance In-Person (Mean of % (SD))	Average of Phone Calls (SD)	Phone Attendance (Mean of % (SD))
Pre-lockdown (14 March 2019–13 March 2020)	Second quarter 2019	13,583	12,144	3.56 (2.75)	0.98 (.07)	1439	3.35 (3.24)	0.64 (0.47)	1.36 (0.86)	0.99 (0.05)
	Third quarter 2019	12,668	11,400	3.16 (2.42)	0.99 (0.06)	1268	2.77 (2.99)	0.56 (0.49)	1.35 (1.08)	0.99 (0.03)
	Fourth quarter 2019	13,325	12,064	3.47 (2.62)	0.99 (0.07)	1261	3.22 (3.18)	0.62 (0.48)	1.37 (0.86)	0.99 (0.06)
	First quarter 2020	13,879	11,252	3.21 (2.34)	0.98 (0.08)	1802	2.95 (2.79)	0.68 (0.46)	1.47 (1.08)	0.99 (0.04)
The first state of alarm and extensions (14 March 2020–21 June 2020)	Second quarter 2020	12,081	6121	2.49 (2.02)	0.98 (0.11)	5960	1.64 (1.58)	0.38 (0.48)	2.77 (2.68)	0.99 (0.05)
Controlled recovery phase (22 June 2020–22 June 2021	Third quarter 2020	12,700	9648	2.87 (2.18)	0.98 (0.07)	3052	2.42 (2.35)	0.59 (0.49)	2.87 (1.66)	0.99 (0.06)
	Fourth quarter 2020	13,907	11,303	3.27 (2.55)	0.99 (0.07)	2604	2.80 (2.74)	0.62 (0.48)	1.66 (1.27)	0.99 (0.06)
	First quarter 2021	15,027	12,505	3.24 (2.53)	0.99 (0.06)	2522	3.00 (3.04)	0.64 (0.48)	1.74 (1.45)	0.99 (0.06)
	Second quarter 2021	15,709	13,437	3.49 (2.66)	0.98 (0.06)	2272	3.35 (3.30)	0.66 (0.47)	1.71 (1.44)	0.99 (0.04)
Restrictions phase-out (22 June 2021–22 June 2022)	Third quarter 2021	13,849	12,019	3.03 (2.31)	0.99 (0.05)	1830	2.89 (2.77)	0.62 (0.48)	1.61 (1.31)	0.99 (0.03)
	Fourth quarter 2021	14,440	12,643	3.22 (2.41)	0.99 (0.06)	1797	3.21 (3.06)	0.64 (0.47)	1.60 (1.21)	0.99 (0.05)
	First quarter 2022	14,880	12,989	3.38 (2.50)	0.99 (0.06)	1891	3.25 (3.11)	0.65 (0.47)	1.61 (1.31)	0.99 (0.06)
	Second quarter 2022	14,802	13,081	3.32 (2.50)	0.99 (0.06)	1721	3.23 (3.03)	0.65 (0.47)	1.60 (1.19)	0.99 (0.05)

**Table 2 healthcare-13-00084-t002:** Comparison between in-person and hybrid modalities of patient characteristics.

	Pre-Lockdown			The First State of Alarm and Extensions			Controlled Recovery Phase			Restrictions Phase-Out		
	In-Person (62.1%)	Hybrid (37.9%)	Chi-Square/Student *t*	In-Person (39.8%)	Hybrid (60.2%)	Chi-Square/Student *t*	In-Person (62.9%)	Hybrid (37.1)	Chi-Square/Student *t*	In-Person (68.9%)	Hybrid (31.1%)	Chi-Square/Student *t*
Years old (M(SD))	39.86 (13.37)	43.97 (10.63)	19.554 **	40.63 (12.96)	43.35 (10.50)	12.680 **	39.46 (13.11)	43.33 (10.34)	24.320 **	39.74 (13.14)	43.55 (10.13)	23.183 **
Male	61.2	38.8	38.505 **	38.5	61.5	40.195 **	61.5	38.5	88.200 **	67.7	32.3	76.320 **
Female	66.1	33.9		45.8	54.2		68.8	31.2		74.0	26.0	
Alcohol	66.2	33.8	77.535 **	43.8	56.3	32.035 **	66.3	33.7	50.162 **	74.0	26.0	135.953 **
Cocaine	64.5	35.5	20.992 **	42.7	57.3	14.604 **	66.8	33.2	60.352 **	72.9	27.1	73.600 **
Opiates	36.9	63.1	2237.643 **^+^	23.1	76.9	712.475 **^+^	36.9	63.1	2528 **^+^	41.9	58.1	3013.009 **^+^
Cannabis	78.1	21.9	469.492 **	56.9	43.1	181.470 **	79.8	20.2	532.343 **	84.7	15.3	544.566 **
Psychotic	47.7	52.3	47.552 **	26.5	73.5	23.384 **	46.9	53.1	61.905 **	51.0	49.0	80.493 **
Mood disorders	49.7	50.3	66.653 **	32.6	67.4	12.997 **	49.5	50.5	76.017 **	56.8	43.2	71.335 **
Anxiety disorders	51.5	48.5	87.315 **	33.3	66.7	19.216 **	52.5	47.5	86.321 **	59.9	40.1	74.838 **
Personality disorders	41.4	58.6	282.567 **	25.3	74.7	87.424 **	43.2	56.8	261.443 **	49.9	50.1	267.977 **
Patients in Mental Health Services	46.5	53.5	218.915 **	27.8	72.7	78.532 **	47.0	53.0	231.567 **	53.2	46.8	243.520 **

** *p* < 0.01; ^+^ Phi > 0.20.

**Table 3 healthcare-13-00084-t003:** Regressions analysis between patients in treatment (value 1) and patients dropping out (value 0).

	Pre-Lockdown	The First State of Alarm and Extensions	Controlled Recovery Phase	Restrictions Phase-Out
Years old	1.014 (1.01–1.02) **	1.012 (1.007–1.016) **	1.015 (1.011–1.018) **	1.011 (1.008–1.014) **
Female	0.980 (0.887–1.081)	1.027 (0.916–1.152)	0.985 (0.904–1.073)	0.973 (0.892–1.061)
Block 1	Cox & Snell’s R^2^ = 0.023 Nagelkerke R^2^ = 0.032	Cox & Snell’s R^2^ = 0.016 Nagelkerke R^2^ = 0.022	Cox & Snell’s R^2^ = 0.024 Nagelkerke R^2^ = 0.034	Cox & Snell’s R^2^ = 0.020 Nagelkerke R^2^ = 0.029
Alcohol	1.097 (0.958–1.258)	1.196 (1.022–1.399) *	1.179 (1.050–1.325) **	1.068 (0.937–1.206)
Cocaine	1.211 (1.060–1.385) **	1.281 (1.097–1.496) **	1.260 (1.123–1.413) **	1.150 (1.021–1.295) *
Opiates	3.844 (3.365–4.393) **	3.916 (3.355–4.570) **	3.995 (3.560–4.482) **	3.338 (2.961–3.764) **
Cannabis	0.993 (0.837–1.177)	1.149 (0.935–1.388)	1.039 (0.899–1.201)	0.933 (0.803–1.083)
Block 2	Cox & Snell’s R^2^ = 0.112 Nagelkerke R^2^ = 0.157	Cox & Snell’s R^2^ = 0.097 Nagelkerke R^2^ = 0.130	Cox & Snell’s R^2^ = 0.111 Nagelkerke R^2^ = 0.158	Cox & Snell’s R^2^ = 0.107 Nagelkerke R^2^ = 0.154
Psychotic	1.459 (1.135–1.875) **	1.340 (1.000–1.797) *	1.285 (1.024–1.614) *	1.354 (1.073–1.709) *
Mood disorders	1.256 (1.055–1.494) *	1.345 (1.097–1.648) **	1.426 (1.213–1.676) **	1.380 (1.172–1.625) **
Anxiety disorders	1.665 (1.448–1.915) **	1.793 (1.523–2.110) **	1.775 (1.565–2.013) **	1.694 (1.493–1.923) **
Personality disorders	1.365 (1.184–1.573) **	1.325 (1.124–1.562) **	1.506 (1.319–1.720) **	1.531 (1.336–1.754) **
Patients in Mental Health Services	1.309 (1.130–1.517) **	1.409 (1.188–1.673) **	1.345 (1.176–1.540) **	1.413 (1.231–1.621)
Block 3	Cox & Snell’s R^2^ = 0.128 Nagelkerke R^2^ = 0.179	Cox & Snell’s R^2^ = 0.116 Nagelkerke R^2^ = 0.156	Cox & Snell’s R^2^ = 0.130 Nagelkerke R^2^ = 0.183	Cox & Snell’s R^2^ = 0.127 Nagelkerke R^2^ = 0.183
Hybrid modality	3.134 (2.906–3.381) **	2.152 (1.188–1.673) **	2.890 (2.702–3.092) **	4.371 (4.081–4.682) **
Block 4	Cox & Snell’s R^2^ = 0.175 Nagelkerke R^2^ = 0.244	Cox & Snell’s R^2^ = 0.140 Nagelkerke R^2^ = 0.187	Cox & Snell’s R^2^ = 0.168 Nagelkerke R^2^ = 0.238	Cox & Snell’s R^2^ = 0.195 Nagelkerke R^2^ = 0.280

* *p* < 0.05; ** *p* < 0 0.01.

## Data Availability

The data are available on request from the author.

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
