# Peer review of "Impact of COVID-19 on Substance Use Disorder Treatment: Examining the Influence of In-Person and Telehealth Intervention on Outcomes Using Real-World Data"

_healthcare, 2025, doi:10.3390/healthcare13010084_

Round 1
Reviewer 1 Report
Comments and Suggestions for Authors
1. At line #79, authors will correct the misspelling of “Desing”.
2. At line #145 to #147, “The analysis of the changes over time revealed that the number of patients remained stable and even increased after the lockdown, when there was a significant decrease” is a confusing expression.
3. At line #184, what are comorbid mental disorders? What are the variables they represented in this study? The authors will clarify them in the Measures subsection; in fact, the authors will expand the Measures section by identifying all variables in this study.
4. At line #224, is "dual pathology" same as "comorbid mental disorders? If not, what are the variables they represented in this study?
5. At line #227 to #228, which table and where in that table showed such indication: “…nearly 40% of 227 these patients receive coordinated care within mental health services”?
Comments on the Quality of English Language
See my comments #1 and #2. The authors will be consistent with their usage of terms (see my comments #2 and #3.
Author Response
- At line #79, authors will correct the misspelling of “Desing”.
Thank you for your comments. The authors have corrected the text.
- At line #145 to #147, “The analysis of the changes over time revealed that the number of patients remained stable and even increased after the lockdown, when there was a significant decrease” is a confusing expression.
The authors have modified the expression. We have proceeded to include the following wording:
“Between the “pre-lockdown” and “restrictions phase out” periods, there was an increase in the number of patients seen. A significant decrease was only observed during the lockdown”.
- At line #184, what are comorbid mental disorders? What are the variables they represented in this study? The authors will clarify them in the Measures subsection; in fact, the authors will expand the Measures section by identifying all variables in this study.
Thank you for your appreciation. We have clarified this concept in line 184:
“The analysis of comorbid mental disorders (any psychotic, mood, anxiety or personality disorders) shows that the presence of comorbid disorders is related to a greater probability of remaining in treatment”
Also, following your recommendation, we have modified the Measure section to include all the variables of the study.
“Second, sociodemographic variables (age, gender), variables associated with substance use (abuse/dependence disorders of alcohol, cocaine, opiates and cannabis use) and variables associated with other comorbid mental disorders (any psychotic disorders, mood disorders, anxiety disorders and personality disorders) and their treatment (patients in mental health services also) were studied as covariates.”
- At line #224, is "dual pathology" same as "comorbid mental disorders? If not, what are the variables they represented in this study?
Thank you for your appreciation. We have used both concepts interchangeably. However, we have proceeded to unify the terminology using the concept “comorbid mental disorders”.
- At line #227 to #228, which table and where in that table showed such indication: “…nearly 40% of 227 these patients receive coordinated care within mental health services”?
Thank you for your comments. We have proceeded to provide further description of the sample in the Participants section. This section indicates that 39.2% of patients with comorbid mental disorders attend mental health services in a coordinated manner:
“The analysis of psychiatric comorbidity shows that 13.7% of the patients had some comorbid mental disorder not induced by drug use (1.8% psychotic disorders, 3.3% mood disorders, 6.1% anxiety disorders and 4.8% personality disorders). Of these patients, 39.2% received health care in a coordinated manner in the mental health services.“
Reviewer 2 Report
Comments and Suggestions for Authors
The manuscript entitled “Impact of COVID-19 on Substance Use Disorder Treatment: Examining the Influence of in-person and telehealth intervention on outcomes using Real-World Data” was interesting. The researchers aimed to analyse the evolution, patient profile and effectiveness of a hybrid care modality (in-person and telehealth) versus an only in-person patient care modality. Major concerns about this manuscript are as follows:
1- The time classification in page 3 needs further attentions, as Covid-19 was identified in December 2019.
2- The effectiveness of the telehealth interventions have not been reported.
3- In the discussion section, the findings need to be supported and compared with other studies.
4- The innovative side of the study is not clear.
5- A conclusion section, should be added.
Comments on the Quality of English LanguageFurther revisions of English Language is required.
Author Response
The manuscript entitled “Impact of COVID-19 on Substance Use Disorder Treatment: Examining the Influence of in-person and telehealth intervention on outcomes using Real-World Data” was interesting. The researchers aimed to analyse the evolution, patient profile and effectiveness of a hybrid care modality (in-person and telehealth) versus an only in-person patient care modality. Major concerns about this manuscript are as follows:
- The time classification in page 3 needs further attentions, as Covid-19 was identified in December 2019.
The authors thank you for your appreciation. We have modified the text to clarify this point:
“Although COVID-19 was identified in December 2019, in many European countries, such as Spain, the legal measures associated with restricting population movements and sanitary services were not adopted until 2020”
- The effectiveness of the telehealth interventions have not been reported.
We thank the reviewer for his appreciation. The authors have contrasted dropout/retention in treatment. This is a commonly used indicator of effectiveness in SUD treatment. However, in order to avoid misunderstandings, the authors have proceeded to eliminate this concept.
- In the discussion section, the findings need to be supported and compared with other studies.
The authors have made a number of changes in this regard. We ask the reviewer to please read this section again.
- The innovative side of the study is not clear.
5- A conclusion section, should be added.
For both questions, the authors believe that the use of real-world data, the number of patients included in the study (44930 outpatients), and the large study period are distinguishing features of this study from previous studies. The authors have proceeded to specify this in the conclusion section requested by the reviewer.
Reviewer 3 Report
Comments and Suggestions for Authors
The purpose of this study was to examine the Impact of COVID-19 on Substance Use Disorder Treatment and analyze the evolution, patient profile, and effectiveness of a hybrid care modality versus an only in-person patient care modality. The authors have presented a well-structured and well-written paper with a clear statement of their hypothesis. However, I would like to offer some comments and suggestions to enhance the clarity and comprehensiveness of the paper.
Abstract
- The title highlights the impact of COVID-19 on SUD treatment; however, the abstract and introduction lack explicit mention of COVID-19. Please ensure alignment between the title, abstract, and manuscript objectives.
2. The term "effectiveness" is used, but its definition remains unclear. Consider explicitly defining what is meant by "effectiveness" in the abstract and throughout the manuscript.
Introduction
- There is insufficient specificity regarding the patient population. Throughout the manuscript, clarify whether references to patients and treatments pertain specifically to individuals with SUD. For example, on page 2, line 46, “Reviews such as that of Mark et al. [8] indicate that the results of in-person and via telehealth treatments are similar.” it is unclear if the statement about telehealth versus in-person care applies to SUD patients or a general patient population.
2. The introduction lacks information about the specific country in which the study was conducted and does not give efficient information about the current situation of SUD patients and treatments and options. It is better to introduce the options that patients have in the country and the availability of telehealth.
3. On page 1, line 36, the term "videoconferencing" is mentioned. Clarify whether this refers to patient care and, if so, provide context regarding its relevance.
4. Clearly specify which outcomes were considered in the study (e.g., clinical outcomes, SUD-related outcomes, or mental health metrics). For example, on page 2, line 66, “Overall, the above studies suggest that, compared with in-person treatment, the use of telehealth techniques lead to results similar to those of in-person treatment, and that the adoption of hybrid treatments may be even more positive.” the outcomes referenced remain ambiguous.
5. Although the intro is well-written to support how telehealth is beneficial these days, the authors should a little elaborate on the disadvantages of telehealth as it has some issues: For instance: 1- not all services can be remote, 2- security and privacy issues that would make patients a bit unconformable, 3- poor Internet connection in some areas, 4- in online services sometimes the medical providers do not have access to the history of patient.
6. Include research questions at the end of the introduction to provide readers with a clear understanding of the study’s focus, particularly given the multiple variables of interest.
Materials and methods
1. Please expand the study design section and write at least a sentence about the study design.
2. In the participants section, provide inclusion and exclusion criteria, including any demographic factors such as age, if applicable.
3. The abstract notes higher retention rates among telehealth users compared to in-person patients. However, given the data collection period during the pandemic, telehealth and hybrid models were likely influenced by pandemic-related social distancing policies, not only medical visits but also everything else was hybrid and Online, like education (Jahanaray, A., Jahanaray, M., & Zohoorian, Z. (2022). Effective factors and issues in online learning in Covid-19: a global review. EDUCATIO: Journal of Education, 7(3), 121-137.), work (Merchant, J. (2021). Working online due to the COVID‐19 pandemic: A research and literature review. Journal of Analytical Psychology, 66(3), 484-505.), mental health utilization (Pasha, A., Qiao, S., Zhang, J., Cai, R., He, B., Yang, X., ... & Li, X. (2024). The impact of the COVID-19 pandemic on mental health care utilization among people living with HIV: A real-world data study. medRxiv.), etc. Consider revising the language to avoid overgeneralizing these findings beyond the pandemic context. This limitation should also be explicitly addressed.
4. Reorganize the methodology section by introducing a subcategory for the data source, followed by sections on study design, participants, and procedures.
5. On page 3, the operationalization of dropout ("six months without treatment") is potentially problematic. Discuss why this criterion was chosen, how follow-up practices typically occur, and whether it is possible that patients discontinued treatment because they no longer required it. Include these concerns as a limitation of the study.
6. “The chi-square test was applied to determine the associations between groups. Given the 131 large sample sizes, to prevent type I errors, the Phi effect size was calculated, considering 132 the presence of at least a weak effect size when phi > .20 [20].” While the use of the Phi coefficient to measure effect size is appropriate for large samples, clarify in the manuscript that Phi does not prevent or control for Type I errors. Instead, adjustments to significance levels or statistical controls should be mentioned. Rephrase the relevant statement on page 3 for accuracy.
7. The sentence "The analysis of the changes over time revealed that the number of patients remained stable and even increased after the lockdown, when there was a significant decrease" is unclear. Please revise for clarity.
Results
1. “The analysis of the changes over time revealed that the number of patients remained stable and even increased after the lockdown, when there was a significant decrease.” The periods mentioned in table require changes in their labels/names. please refer to any valid sources or even https://www.worldometers.info your last period called "New Normality " is just in the biggest outbreak in Spain. You can call these period as different pandemic periods but all are in the pandemic.
2. Clarify the duration of the lockdown in Spain mentioned in Table 1 (e.g., "four months"). Include appropriate references to validate this information.
3. On page 6, the role of socio-demographic variables is mentioned “The socio-demographic variables have scarce predictive capacity for treatment maintenance or dropout.” If these variables were considered covariates in the regression model, please specify this in the methods section.
4. On page 6, line 181, where the R² values are discussed, include the exact statistics to aid readers who may find the table challenging to interpret. Ensure that the narrative is a clear representation of the results.
Discussion
- On page 7, line 201, the discussion reiterates results rather than interpreting them. Instead of restating findings, elaborate on their implications and potential explanations. For example, why might hybrid care result in improved retention?
2. “This finding is even more relevant since patients in hybrid care have a lower attend-237 ance rate in drug treatment centres. Therefore, this result shows that telehealth care activ-238 ity in combination with in-person care could improve patient retention. Moreover, the probability of patients receiving hybrid care was observed during all phases of the study. In that sense, this result extends the evidence for the benefits of hybrid care beyond the periods directly affected by the COVID-19 pandemic. Despite the favourable evidence supporting hybrid” Please try having some reference and support for your findings instead of self-reflection on the study's results.
3. Emphasize the study's implications and suggest how these findings could inform policy, clinical practice, or future research. For instance, what recommendations can be made for optimizing hybrid care? Also, provide future directions based on the study's limitations.
Overall, the authors have produced an insightful paper that illuminates an important topic. By implementing these suggestions, the paper can be further refined and contribute to the existing body of research in this area.
Author Response
The purpose of this study was to examine the Impact of COVID-19 on Substance Use Disorder Treatment and analyze the evolution, patient profile, and effectiveness of a hybrid care modality versus an only in-person patient care modality. The authors have presented a well-structured and well-written paper with a clear statement of their hypothesis. However, I would like to offer some comments and suggestions to enhance the clarity and comprehensiveness of the paper.
Abstract
- The title highlights the impact of COVID-19 on SUD treatment; however, the abstract and introduction lack explicit mention of COVID-19. Please ensure alignment between the title, abstract, and manuscript objectives.
We appreciate your suggestion. The authors have revised the abstract and the introduction to make it explicit that the focus of the study is on the period during which different measures associated with the COVID pandemic have been adopted.
- The term "effectiveness" is used, but its definition remains unclear. Consider explicitly defining what is meant by "effectiveness" in the abstract and throughout the manuscript.
The authors have considered eliminating this term and have made the objectives of the study more explicit.
Introduction
- There is insufficient specificity regarding the patient population. Throughout the manuscript, clarify whether references to patients and treatments pertain specifically to individuals with SUD. For example, on page 2, line 46, “Reviews such as that of Mark et al. [8] indicate that the results of in-person and via telehealth treatments are similar.” it is unclear if the statement about telehealth versus in-person care applies to SUD patients or a general patient population.
We thank you for your appreciation. The authors have proceeded to specify that the studies cited were performed in SUD patients:
“Different studies have been conducted during specific phases of the COVID-19 pandemic with USD patients. For example, reviews such”
- The introduction lacks information about the specific country in which the study was conducted and does not give efficient information about the current situation of SUD patients and treatments and options. It is better to introduce the options that patients have in the country and the availability of telehealth.
We thank the reviewer for his appreciation. In the description of the participants we have included a paragraph indicating the options patients had for telemedicine:
“During these phases, all patients were informed that appointments could be either in-person or conducted through telehealth modalities, including computerized assessments, telephone-based recovery and telephone-based therapy. Patients could accept or decline treatment including telehealth modalities. They could also indicate which telehealth modality they preferred at any given time and could make use of these at any time during treatment.”
- On page 1, line 36, the term "videoconferencing" is mentioned. Clarify whether this refers to patient care and, if so, provide context regarding its relevance.
Following the reviewer's suggestion, the authors have proceeded to clarify the term:
“…videoconferencing between therapists and patients in routine clinical practice,…”
- Clearly specify which outcomes were considered in the study (e.g., clinical outcomes, SUD-related outcomes, or mental health metrics). For example, on page 2, line 66, “Overall, the above studies suggest that, compared with in-person treatment, the use of telehealth techniques lead to results similar to those of in-person treatment, and that the adoption of hybrid treatments may be even more positive.” the outcomes referenced remain ambiguous.
The authors have proceeded to specify the outcomes in which similar results are found:
“Overall, the above studies suggest that, compared with in-person treatment, the use of telehealth techniques lead to results similar to those of in-person treatment on indicators associated with dropout, retention in treatment or quality of life, and that the adoption of hybrid treatments may be even more positive”
- Although the intro is well-written to support how telehealth is beneficial these days, the authors should a little elaborate on the disadvantages of telehealth as it has some issues: For instance: 1- not all services can be remote, 2- security and privacy issues that would make patients a bit unconformable, 3- poor Internet connection in some areas, 4- in online services sometimes the medical providers do not have access to the history of patient.
Following the reviewer's suggestions, we have incorporated the limitations identified for telehealth:
“However, it has also been noted that this modality makes patient-therapist interaction difficult, which can be a handicap for treatment [7]. Further, not all treatment services can be remotely implemented; security and privacy issues may cause patients to feel a bit uncomfortable; restricted internet access in some areas; or limited availability to patient’s medical history by healthcare providers. This means that telehealth services cannot be provided on an equal basis to all patients.”
- Include research questions at the end of the introduction to provide readers with a clear understanding of the study’s focus, particularly given the multiple variables of interest.
Following the recommendations of the reviewer, the authors have modified the end of the introduction, adding a research question:
“However, most of these studies were conducted during specific time periods, and their findings may have been influenced by the temporary legal measures adopted by governments to mitigate the impact of the COVID-19 pandemic. In turn, all of the above studies have been conducted in the US. Therefore, it can be useful to provide evidence over an extended period, in countries other than US, to evaluate how telehealth services have been implemented in addiction treatment centers. Specifically, how has telehealth services evolved from the restrictions imposed by the COVID-19 pandemic to their lifting, compared to face-to-face care? Additionally, what impact do they have on patient retention?”
Materials and methods
- Please expand the study design section and write at least a sentence about the study design.
Following the reviewer's recommendations, the authors have explained the study design in more detail:
“Retrospective observational study based on data from patients receiving treatment between 03/14/2019 and 06/21/2022. A follow-up period of one year was included for these patients until 06/22/2023.”
- In the participants section, provide inclusion and exclusion criteria, including any demographic factors such as age, if applicable.
The authors have proceeded to make the inclusion and exclusion criteria more explicit. There were no socio-demographic variables that led to exclusion from the study.
- The abstract notes higher retention rates among telehealth users compared to in-person patients. However, given the data collection period during the pandemic, telehealth and hybrid models were likely influenced by pandemic-related social distancing policies, not only medical visits but also everything else was hybrid and Online, like education (Jahanaray, A., Jahanaray, M., & Zohoorian, Z. (2022). Effective factors and issues in online learning in Covid-19: a global review. EDUCATIO: Journal of Education, 7(3), 121-137.), work (Merchant, J. (2021). Working online due to the COVID‐19 pandemic: A research and literature review. Journal of Analytical Psychology, 66(3), 484-505.), mental health utilization (Pasha, A., Qiao, S., Zhang, J., Cai, R., He, B., Yang, X., ... & Li, X. (2024). The impact of the COVID-19 pandemic on mental health care utilization among people living with HIV: A real-world data study. medRxiv.), Consider revising the language to avoid overgeneralizing these findings beyond the pandemic context. This limitation should also be explicitly addressed.
Following the reviewer's recommendations, we have incorporated these observations both in the abstract of the MS and in discussion of the study:
“Therefore, in line with previous review studies [35, 36], this result shows that telehealth care activity in combination with in-person care could improve patient retention. However, before considering telehealth as one of the most important factors in improving the retention of SUD patients, several factors should be taken into account. In this context, it is likely that factors such as educational level, employment, or availability of mental health facilities [37-39] influenced the use of a hybrid or in-person model. Therefore, it is likely that there is an interplay of personal factors combined with telehealth measures, improve retention [40].”
- Reorganize the methodology section by introducing a subcategory for the data source, followed by sections on study design, participants, and procedures.
Following the reviewer's suggestions, we have included a first section on the data source. Much of the information had been described in the ‘procedure’ section, so we have removed that subcategory.
- On page 3, the operationalization of dropout ("six months without treatment") is potentially problematic. Discuss why this criterion was chosen, how follow-up practices typically occur, and whether it is possible that patients discontinued treatment because they no longer required it. Include these concerns as a limitation of the study.
We agree with the reviewer that it is complex to determine a time period for considering treatment abandonment, as there are numerous criteria (Narvaez-Camargo, et al., submitted). Therefore, we have followed the reviewer's suggestion and have included it as an limitation:
“On the other hand, there are different criteria for determining when a patient has dropped out of treatment or remains in treatment (41, 42). In this study we have chosen to define treatment drop-out after a period of six months without attending appointments before clinical recommendation. During this period of time, patients may feel well and believe that they have overcome their addiction problems and could therefore stop treatment. It may therefore be appropriate to combine this outcome indicator with other quality of life measures. Nevertheless, studies consistently show that patients who dropout of treatment before medical recommendations generally require more therapeutic support due to relapses [43, 44].”
Narvaez-Camargo, M., Lozano, O. M., Mancheño-Velasco, C., Vedejo-García, A. (Submitted). Substance use disorder treatment outcome: methodological overview of outcomes. International Journal of Methods in Psychiatry Research.
- “The chi-square test was applied to determine the associations between groups. Given the large sample sizes, to prevent type I errors, the Phi effect size was calculated, considering the presence of at least a weak effect size when phi > .20 [20].” While the use of the Phi coefficient to measure effect size is appropriate for large samples, clarify in the manuscript that Phi does not prevent or control for Type I errors. Instead, adjustments to significance levels or statistical controls should be mentioned. Rephrase the relevant statement on page 3 for accuracy.
The authors agree with the reviewer, and we have proceeded to delete the reference to the type I error.
- The sentence "The analysis of the changes over time revealed that the number of patients remained stable and even increased after the lockdown, when there was a significant decrease" is unclear. Please revise for clarity.
The authors have modified the expression. We have proceeded to include the following wording:
“Between the “pre-lockdown” and “restrictions phase out” periods, there was an increase in the number of patients seen. A significant decrease was only observed during the “first state of alarm and extensions” period.”
Results
- “The analysis of the changes over time revealed that the number of patients remained stable and even increased after the lockdown, when there was a significant decrease.” The periods mentioned in table require changes in their labels/names. please refer to any valid sources or even https://www.worldometers.info your last period called "New Normality " is just in the biggest outbreak in Spain. You can call these period as different pandemic periods but all are in the pandemic.
The measures adopted by governments have had different terminology and time periods depending on the evolution of the COVID pandemic and the vaccination of the population. The authors have proceeded to incorporate the laws in Spain that have had the greatest impact on the care of patients with SUD (although they are not the only ones). We have renamed the different phases)
- Clarify the duration of the lockdown in Spain mentioned in Table 1 (e.g., "four months"). Include appropriate references to validate this information.
The authors have specified the periods of analysis in the table and incorporated references to the most relevant legal health measures.
- On page 6, the role of socio-demographic variables is mentioned “The socio-demographic variables have scarce predictive capacity for treatment maintenance or dropout.” If these variables were considered covariates in the regression model, please specify this in the methods section.
Following the suggestions of another reviewer, the variables used in the study have been included in the ‘measure’ section. It has been specified that they have been used as covariates. The authors hope that you will find it appropriate to specify them in this sub-section.
- On page 6, line 181, where the R² values are discussed, include the exact statistics to aid readers who may find the table challenging to interpret. Ensure that the narrative is a clear representation of the results.
Following the reviewer's suggestion, the authors have included the exact value
Discussion
- On page 7, line 201, the discussion reiterates results rather than interpreting them. Instead of restating findings, elaborate on their implications and potential explanations. For example, why might hybrid care result in improved retention?
- “This finding is even more relevant since patients in hybrid care have a lower attend-ance rate in drug treatment centres. Therefore, this result shows that telehealth care activity in combination with in-person care could improve patient retention. Moreover, the probability of patients receiving hybrid care was observed during all phases of the study. In that sense, this result extends the evidence for the benefits of hybrid care beyond the periods directly affected by the COVID-19 pandemic. Despite the favourable evidence supporting hybrid” Please try having some reference and support for your findings instead of self-reflection on the study's results.
- Emphasize the study's implications and suggest how these findings could inform policy, clinical practice, or future research. For instance, what recommendations can be made for optimizing hybrid care? Also, provide future directions based on the study's limitations.
The authors have tried to respond to these two suggestions. We ask the reviewer to please read this section again.
Round 2
Reviewer 1 Report
Comments and Suggestions for Authors
The revisions are appropriate and valid.
Author Response
Comments 1: The revisions are appropriate and valid.
Response: Thank you very much for the suggestions you made in the first round.
Reviewer 2 Report
Comments and Suggestions for Authors
Thanks for revising the manuscript. In the methods section, "data source" can be reported as 2.3section, after reserach design and participants.
Author Response
Comments 1: Thanks for revising the manuscript. In the methods section, "data source" can be reported as 2.3section, after reserach design and participants.
Response: We appreciate your suggestion. This subsection was included following the suggestions of another reviewer who pointed us to it: "Reorganize the methodology section by introducing a subcategory for the data source, followed by sections on study design, participants, and procedures"